https://doi.org/10.1038/s41467-022-28519-x | OPEN

# Charge-noise spectroscopy of Si/SiGe quantum dots via dynamically-decoupled exchange oscillations

Elliot J. Connors [1], J. Nelson[1], Lisa F. Edge[2] & John M. Nichol [1✉]

Electron spins in silicon quantum dots are promising qubits due to their long coherence times, scalable fabrication, and potential for all-electrical control. However, charge noise in the host semiconductor presents a major obstacle to achieving high-fidelity single- and two-qubit gates in these devices. In this work, we measure the charge-noise spectrum of a Si/SiGe singlet-triplet qubit over nearly 12 decades in frequency using a combination of methods, including dynamically-decoupled exchange oscillations with up to 512 $\pi$ pulses during the qubit evolution. The charge noise is colored across the entire frequency range of our measurements, although the spectral exponent changes with frequency. Moreover, the charge-noise spectrum inferred from conductance measurements of a proximal sensor quantum dot agrees with that inferred from coherent oscillations of the singlet-triplet qubit, suggesting that simple transport measurements can accurately characterize the charge noise over a wide frequency range in Si/SiGe quantum dots.

[1] Department of Physics and Astronomy, University of Rochester, Rochester, NY 14627, USA. [2] HRL Laboratories LLC, 3011 Malibu Canyon Road, Malibu, CA 90265, USA. ✉email: john.nichol@rochester.edu

Despite their long coherence times, which lead to high-fidelity single-[1–3] and two-qubit gates[4–7], electron spins in Si/SiGe quantum dots suffer from charge noise. Most single- and two-qubit gates rely on manipulating electrons by precisely controlling their local electrostatic potentials, which typically result from voltages applied to gate electrodes, but are also affected by charge fluctuations in the environment. As a result, charge noise in the semiconductor environment often limits the fidelity of these operations[2,5]. A picture has emerged of an approximately, but not exactly, $1/f$-like charge-noise spectrum that may originate from surfaces or interfaces in Si/SiGe and other Si devices[5,8–13]. Questions remain, however, about the relationship between the low- and high-frequency parts of the spectrum, how the noise can be mitigated, and what, if any quantum operations can be designed[5,8–13], to mitigate the charge noise. Moreover, despite numerous theoretical predictions about different possible reasons for charge noise, the lack of sufficient experimental data prevents confirming or denying these predictions. Because of the critical importance of charge noise in relation to spin-qubit gate fidelities, further measurements to help elucidate methods to eliminate or circumvent it are essential for the continued progress of Si spin qubits.

In this work, we report noise-spectroscopy of Si/SiGe quantum dots over 12 decades in frequency from near 1 μHz to above 1 MHz using a suite of measurement techniques, including dynamically-decoupled exchange oscillations. Previous methods for measuring MHz-level charge fluctuations in quantum dots have relied on dynamic nuclear polarization[14,15], a micromagnet[5], or an antenna for high-frequency spin manipulation[9]. The approach we present here requires two electrons in a double quantum dot and a difference in the longitudinal Zeeman energy between the dots. Here, we use only a uniform in-plane external field and the naturally-occurring $g$-factor difference between the dots[16–18] to generate this difference without an external gradient source. The noise spectrum we observe is approximately described by a power law over the entire frequency range, although deviations in the spectral exponent provide further evidence that the noise may originate from an inhomogeneous distribution of two-level systems[11].

## Results

**Device description**. The device we use is fabricated on an undoped Si/SiGe heterostructure with a strained quantum well made of natural Si ~50 nm below the surface. An overlapping-gate architecture[19,20] allows precise control over the electronic confinement in the underlying two-dimensional electron gas (Fig. 1a, b). For all measurements reported here, we use a double quantum dot beneath plunger gates $P_1$ and $P_2$, and a sensor dot underneath S. We use rf reflectometry[21,22] to measure the conductance of the sensor quantum dot on microsecond timescales. The device is cooled in a dilution refrigerator to a base temperature of ~50 mK, and we apply an in-plane magnetic field $B_{ext} = 500$ mT.

We operate the double dot at the (3,1)–(4,0) transition, with 3(1) or 4(0) electrons in dot 1(2). In the ground state configuration, the two lowest-energy electrons in dot 1 occupy the lowest-energy valley level as a singlet and do not appreciably contribute to the dynamics of the other electrons. The singlet-triplet splitting in the (4,0) configuration is limited by the orbital energy spacing, rather than the smaller valley splitting[21,23–25]. This large singlet-triplet splitting facilitates operation of the double dot as a singlet-triplet (S-T$_0$) qubit. In the $\{|S\rangle, |T_0\rangle\}$ basis, where $|S\rangle = \frac{1}{\sqrt{2}}(|\uparrow\downarrow\rangle - |\downarrow\uparrow\rangle)$ and $|T_0\rangle = \frac{1}{\sqrt{2}}(|\uparrow\downarrow\rangle + |\downarrow\uparrow\rangle)$, the effective S-T$_0$ qubit Hamiltonian is given by $H_{ST} = J(\epsilon, V_T) S_z + \Delta B_z S_x$[26]. $J(\epsilon, V_T)$ is the exchange coupling, which depends

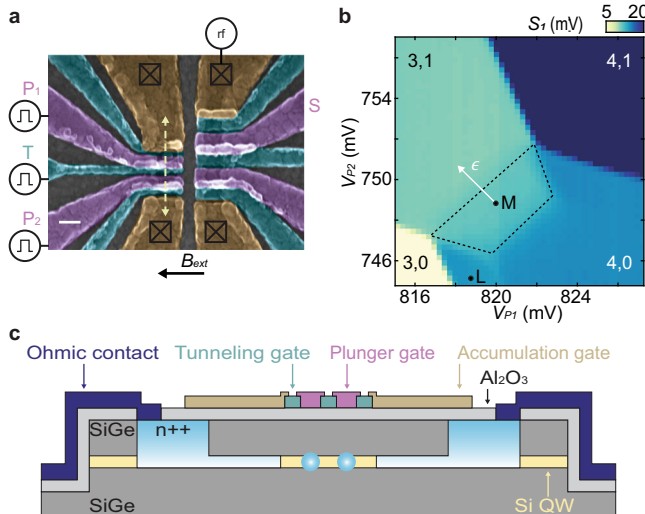

**Fig. 1 Experimental setup. a** False-color scanning electron micrograph of a device nominally identical to the one tested. The white scale bar represents 100 nm. The S-T$_0$ qubit is formed under plunger gates P$_1$ and P$_2$. The sensor quantum dot is formed under gate S. **b** Pauli spin blockade at the (3,1)-(4,0) charge transition. The trapezoid in the (4,0) charge configuration indicates the spin blockade region. $S_1$ is the measured charge-sensor signal. Positions L and M are the singlet load and the measure positions, respectively. Position M also serves as the idle position of the plunger gate dc voltages and defines the position $\epsilon = 0$. **c** Cross section of the device along the dashed arrow in **a**.

on the double-dot detuning $\epsilon$, and $V_T$, the voltage pulse applied to the barrier gate T[27,28]. We define $\epsilon = 0$ at the measurement point (Fig. 1b). $\Delta B_z$ is the difference in the longitudinal Zeeman splitting at the location of the two dots, which we believe originates primarily from electron $g$-factor differences between the two dots[16–18] (see Supplementary Note 3). $S_x$ and $S_z$ are spin 1/2 operators in the $\{|S\rangle, |T_0\rangle\}$ basis.

Dynamically-decoupled exchange oscillations in S-T$_0$ qubits, which are useful for measuring high-frequency charge noise[14], usually involve applying $X$ gates to refocus exchange oscillations. Previously, $X$ gates have been realized via free evolution when $\Delta B_z \gg J$, which can be achieved through dynamic nuclear polarization or through the use of micromagnets. In the present device, when $B_{ext} = 500$ mT, $\Delta B_z \approx 3.85 \pm 0.25$ MHZ (see Supplementary Note 3). When $V_T = 0$, we find that the minimum value of exchange coupling in our device is between 0.5 and 2.1 MHz, and we are not able to achieve $\Delta B_z \gg J$. To circumvent this problem, we generate a Hadamard gate $H$ by tuning $\epsilon$ such that $J(\epsilon, V_T) = \Delta B_z$ (see Supplementary Note 1) and synthesize a composite $X$ gate as $HZH$, where $Z$ indicates a $\pi$ rotation around the $z$ axis, generated via free evolution under exchange[29]. The maximum value of exchange coupling in this device is larger than 100 MHz, so we can easily achieve $J \gg \Delta B_z$ as required for a $Z$ gate. As discussed further below, this composite $X$ gate enables dynamically-decoupled exchange oscillations with up to 512 $\pi$ pulses during coherent evolution. The Hadamard operation may thus be a useful tool for future efforts to entangle S-T$_0$ qubits[30]. We also advantageously use the Hadamard gate to prepare superposition states of the S-T$_0$ qubit.

**Free-induction decay measurements**. We first investigate charge noise through exchange-driven free-induction decay (FID) experiments. After preparing the S-T$_0$ qubit in a superposition state by applying an $H$ gate to an initialized singlet state, we pulse $\epsilon$ and $V_T$ to generate $J \gg \Delta B_z$. The S-T$_0$ qubit evolves under $J$ for a

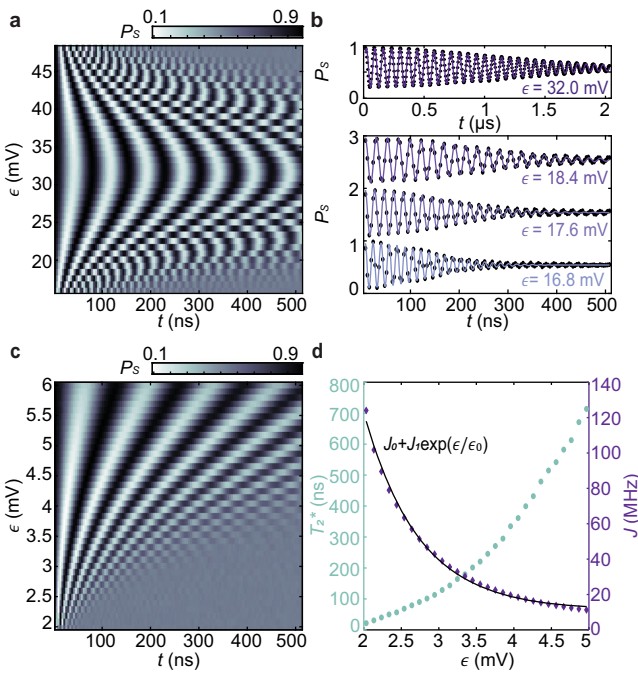

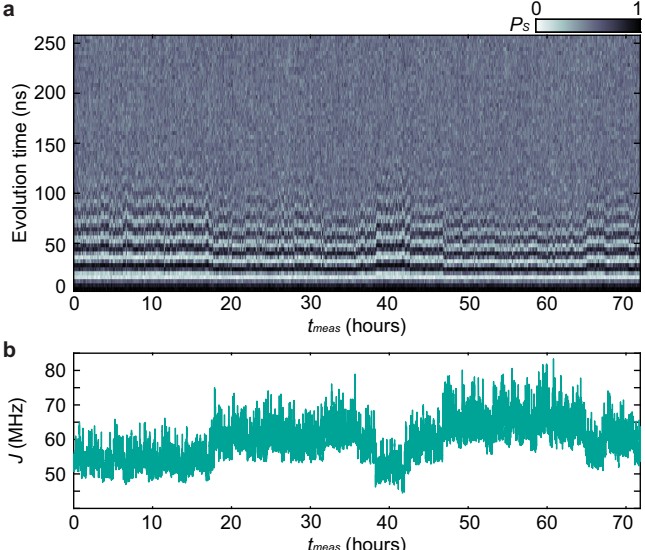

**Fig. 2 FID experiments. a** Exchange oscillations of the S-$T_0$ qubit with $V_T = 90$ mV and varying $\epsilon$. **b** Top panel: exchange oscillations in the symmetric configuration with $\epsilon = 32$ mV. Bottom panel: individual lines from the data shown in **a** and corresponding fits. The data are offset for visibility. **c** Exchange oscillations at small $\epsilon$ demonstrating detuning-control ($V_T = 0$ mV). **d** Values of $T_2^*$ (left axis) and $J$ (right axis) as a function of $\epsilon$ extracted from the data shown in **c**. The black line is a fit of the measured values of $J$ from **c** between $\epsilon = 2$ mV and $\epsilon = 5$ mV to a function $J(\epsilon) = J_0 + J_1 \exp(\epsilon/\epsilon_0)$, from which we extract $dJ/d\epsilon$.

variable time $t$. We then apply another $H$ gate, followed by Pauli spin-blockade readout. We observe both tilt- and barrier-controlled exchange oscillations as shown in Fig. 2a–c.

The dephasing of exchange oscillations primarily results from electrochemical potential noise, which induces detuning fluctuations. In our device, $|dJ/d\epsilon|$ is approximately one order of magnitude larger than $|dJ/dV_T|$. Thus we neglect fluctuations in $V_T$ in the following analysis. We observe that as $\epsilon$ increases and $J$ decreases, $|dJ/d\epsilon|$ also decreases, and the coherence time $T_2^*$ increases. As expected, the oscillation amplitude decays as $\exp[(-t/T_2^*)^2]$, which is approximately consistent with $1/f$ noise[31].

We obtain $T_2^*$ and $J(\epsilon)$ by fitting the exchange oscillations for each value of $\epsilon$ (Fig. 2b, d). We extract $dJ/d\epsilon$ by fitting $J(\epsilon)$ to a smooth function and differentiating it (Fig. 2d). Assuming a single-sided $1/f$ noise spectrum for the electrochemical potential of each quantum dot $S_\mu(f) = A_\mu/f$, we estimate $A_\mu = 0.42$ μeV$^2$ from the extracted values of $T_2^*$ and $dJ/d\epsilon$[14] and compute an RMS electrochemical potential noise $\sigma_\mu = 2.7$ μeV (see Methods). Extracting $A_\mu$ and $\sigma_\mu$ via FID measurements in this way assumes that the noise is described across the relevant frequency range by a power-law with a single spectral exponent. However, refs. [11,13,32] have shown that this assumption does not always hold. Thus, neither $A_\mu$ nor $\sigma_\mu$ completely characterize the noise. In extracting $A_\mu$ and $\sigma_\mu$, we have assumed that the electrochemical potentials of neighboring dots fluctuate independently. This assumption is supported by temporal correlation measurements described in Methods.

To precisely measure the low-frequency noise spectrum, we repeatedly measure exchange oscillations approximately once

every second for 71.81 h, resulting in a total of 262,144 exchange-oscillation traces (Fig 3a)[13]. We then fit each trace to extract the oscillation frequency as a function of measurement time, $J(t_{meas})$, which we finally convert to a signal $\epsilon(t_{meas})$ via our fit of $J(\epsilon)$ (Fig 3b). Because the effective sampling rate ($\sim$1 Hz) is not perfectly constant over the entire 3-day FID experiment (see Methods), we calculate the corresponding spectrum of the time series $\epsilon(t_{meas})$ via a combination of Bartlett's[33] and the Lomb–Scargle[34] methods. Bartlett's method reduces the variance of the acquired spectrum by averaging spectra calculated from $N_W$ non-overlapping windows of data together, although the minimum visible frequency increases with the number of windows used. Using $N_W = 100$, 30, 10, 3, and 1, we map the charge-noise spectrum from $f = 3.9$ μHz to $\sim$0.5 Hz. These measurements provide a more detailed picture of the low-frequency noise than the estimations made from the dephasing of exchange oscillations discussed above. In particular, these measurements indicate that the noise is not described by a single spectral exponent across the relevant frequency range.

**Dynamical-decoupling measurements.** Having investigated the low-frequency charge noise, we turn our attention to the high-frequency charge noise. Dynamical-decoupling schemes, which suppress the effects of low-frequency noise, are useful for investigating high-frequency noise sources[5,14,35–37]. In particular, the Carr-Purcell-Meiboom-Gill (CPMG) sequence[31,35], which uses multiple $\pi$ pulses to refocus inhomogeneous broadening, can enable high-precision noise spectroscopy. Here, we perform exchange-based CPMG measurements using the S-$T_0$ qubit by refocusing exchange oscillations with $1 \leq n_\pi \leq 512$ composite $X$ gates spaced by a time interval $t_{int}$ during the evolution. Each CPMG experiment is characterized by the total evolution time, $\tau = n_\pi t_{int}$, the finite duration of each $\pi$ pulse, $t_\pi$, and the total number of refocusing pulses, $n_\pi$. The total sequence time is given by $t = \tau + n_\pi t_\pi$. A schematic of the pulse sequence of an exchange-CPMG experiment is shown in Fig. 4a. For a given set of $\tau$, $t_\pi$, and $n_\pi$, the spectral weighting function, $W(f; \tau, t_\pi, n_\pi)$, describes the qubit-noise coupling in the frequency domain

**Fig. 3 Three-day FID experiment. a** Exchange oscillations as a function of $t_{meas}$. A total of 262,144 oscillation traces are acquired during the measurement time. **b** Plot of $J$ as a function of $t_{meas}$. $J(t_{meas})$ is extracted from the data shown in **a** by fitting each trace to a function $P_S(t) = [A\cos(2\pi Jt) + B\sin(2\pi Jt)]\exp\left[-(t/T_2^*)^2\right] + c$ with $A$, $B$, $J$, $T_2^*$, and $c$ as fit parameters.

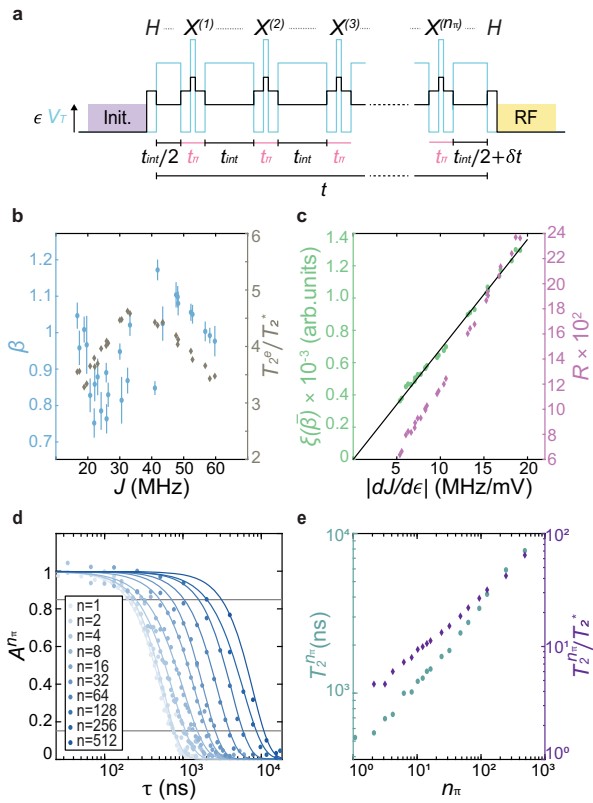

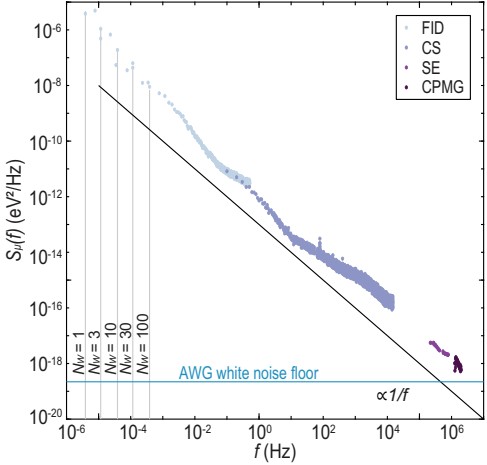

**Fig. 5 Charge-noise spectrum.** Plot of the single-sided charge-noise spectrum determined from a combination of FID, charge-sensor (CS), spin-echo (SE), and CPMG experiments. Data points around 60 Hz and multiples thereof have been omitted. A black trendline proportional to $f^{-1}$ is shown. Spin-echo measurements are plotted at frequencies that maximize $W(f; T_2^e, t_\pi, 1)$. Gray vertical lines indicate the minimum resolvable frequency for the different values of $N_W$. A blue line indicates the estimated white noise floor from the arbitrary waveform generator (AWG), which is approximately two orders of magnitude larger than the estimated Johnson–Nyquist noise from other resistors in our experimental setup.

**Fig. 4 CPMG experiments. a** Schematic of the CPMG pulse sequence. The qubit is prepared along the x-axis and evolves under exchange for a total time $\tau + \delta t$, during which we periodically apply composite X gates. **b** $\beta$ versus $J$ extracted from fits of the echo amplitude decay (left axis) and $T_2^e/T_2^*$ (right axis) versus $J$. Error bars represent the standard error of the fit for $\beta$. **c** Plot of $\xi(\bar{\beta})$ (left axis) and $R$ (right axis) versus $|dJ/d\epsilon|$. The $\xi(\bar{\beta})$ data are well fit to a straight line with a y-intercept of zero (black line), which is consistent with a charge-noise spectrum of the form $S = A/f^\beta$. **d** Plot of the normalized amplitude of the CPMG oscillations as a function of $\tau$ for select CPMG experiments. **e** $T_2^{n_\pi}$ (left axis) and the ratio $T_2^{n_\pi}/T_2^*$ (right axis) as a function of $n_\pi$.

during the CPMG sequence (see Methods). We measure the high-frequency portion of the qubit-charge-noise spectrum by performing CPMG experiments with different $n_\pi$.

We first implement CPMG with $n_\pi = 1$, which is equivalent to a spin-echo experiment[5,14,37,38], at various values of $\epsilon$ corresponding to evolution at different values of $J$ ranging from ~15 to 60 MHz. We analyze our data within the filter-function formalism, taking into account the finite duration of the $\pi$ pulse[39]. A detailed description of the analysis of all CPMG measurements, including spin-echo measurements, is given in Methods. For each spin-echo measurement, we assume noise with a spectral form $S_\mu(f) = A_\mu/f^\beta$ and extract values of $\beta$ as well as the echo coherence time, $T_2^e$, from a fit of the amplitude decay as a function of $\tau$. ($T_2^e$ is the value of $\tau$ when the amplitude decay envelope diminishes to $1/e$ times its initial value.) We find an average value of $\bar{\beta} = 0.95 \pm 0.12$, and an improvement in the coherence time of a factor of approximately four across our measurements (Fig. 4b). Furthermore, for a power-law noise spectrum we expect $\xi(\beta) \equiv (T_2^e + t_\pi)^{-(\beta+1)/2}$ to be proportional to $dJ/d\epsilon$ when $R = t_\pi/t$, the ratio of the $\pi$-pulse time to the total sequence time, is a small number. Figure 4c shows plots of $\xi(\bar{\beta})$ and $R$ versus $dJ/d\epsilon$ demonstrating good agreement with this

expectation. Together, these data suggest that the charge-noise spectrum can be described by a power law with a single spectral exponent $\bar{\beta}$ for frequencies near $1/T_2^e$. Last, from the extracted values of $T_2^e$, $\beta$, and $dJ/d\epsilon$, we estimate a value of the noise power spectrum at frequencies where $W(f; T_2^e, t_\pi, 1)$ is a maximum for each individual experiment (Fig. 5).

We also perform CPMG experiments with $1 < n_\pi \leq 512$ at evolution frequencies $J = 50 \pm 2$ MHz. Figure 4d shows the normalized amplitude, $A^{n_\pi}$, for $n_\pi = 2, 4, 8, 16, ..., 512$. For CPMG experiments with large $n_\pi$ and a not-too-small duty-cycle, $D = \tau/t$ (see Methods), the spectral weighting function can be approximated as a $\delta$-function at a frequency $f = n_\pi D/(2\tau)$. Thus, for CPMG experiments with $n_\pi \geq 8$ and $D \geq 0.2$, we calculate the single-sided spectrum as

$$S_\mu\left(f = \frac{n_\pi D}{2\tau}\right) \simeq -\frac{\ln(A^{n_\pi})}{\pi^2 \tau D}\left(\frac{dJ}{d\epsilon}\right)^{-2}\left(\alpha_{P1}^{-2} + \alpha_{P2}^{-2}\right)^{-1} \quad (1)$$

for each of the data points within the range $0.15 < A^{n_\pi} < 0.85$[5]. Here, $\alpha_{P1(2)}$ is the lever arm of gate $P_{1(2)}$ in units of eV/V. We describe the process of extracting noise spectra from CPMG experiments with finite duration $\pi$-pulses, and the expected error in this process, in detail in Methods.

**Charge-sensor measurements.** To fill in the frequency range between our low- and high-frequency measurement techniques, we measure the sensor-dot charge-noise spectrum. With the sensor dot tuned such that its conductance is sensitive to fluctuations in the electrochemical potential, and with the detuning of the S-$T_0$ qubit set near the center of the (3,1) charge region, we sample the reflected rf signal at 100 kHz for ~500 s. We calculate the power spectrum of the acquired signal using Bartlett's method with 50 windows (Fig. 5). The high bandwidth of the rf-reflectometry circuit enables measurement of the charge-noise spectrum to about 10 kHz. From a power-law fit of the acquired spectrum from $f = 0.1$ to 10 Hz, we extract a spectral density $S_\mu(1 \text{ Hz}) = 0.81 \pm 0.02 \ \mu\text{V}^2/\text{Hz}$.

We have observed that the spectra acquired using this approach can depend on the value of $\epsilon$ (see Methods). Specifically, when $\epsilon \approx 0$, we observe a strong Lorentzian component in the acquired spectrum at frequencies of ~10–1000 Hz. This feature is minimized when $\epsilon$ is large such that the electrons are deep in the (3,1) configuration. We speculate that this noise feature results from electrons hopping between dots, or from electrons hopping between the dots and reservoirs. Both mechanisms are expected to be more pronounced when $\epsilon$ is near the (3,1)–(4,0) transition. Increasing the amplitude of the rf carrier has a similar effect (see Methods). The rf carrier may itself induce the suspected charge-hopping and/or electron-exchange events. It is possible, though unlikely, that the exchange-oscillation measurements discussed above are also affected by the reflectometry signal, which we use to measure the S-$T_0$ qubit. The reflectometry signal is unblanked only during the readout portion of typical experiments, and only for microsecond-scale durations.

**Charge-noise spectrum**. Figure 5 shows all of the charge-noise measurements discussed above. In total, our measurements span a frequency band from 4 µHz to 2 MHz. As discussed above, noise spectra calculated from the FID data using $N_W = 100, 30, 10, 3,$ and 1 have minimum visible frequencies of ~387, 116, 39, 12, and 4 µHz, respectively. We plot each of the resulting spectra only from their minimum visible frequency to the minimum visible frequency of the spectrum corresponding to the next largest $N_W$. For $N_W = 100$, we plot the spectrum to 15 kHz. Above this frequency, the spectrum is comparable to or less than the estimated noise floor (see Methods). We have also removed data in 10 Hz windows centered around integer multiples of 60 Hz from 60 to 1920 Hz (see Methods). Values of the noise spectra acquired from echo measurements are plotted at frequencies where $W(f; T_2^e, t_\pi, 1)$ is a maximum, and thus where the individual experiments are most sensitive to noise.

We comment on several aspects of our data. First, despite the wide frequency range, the different measurements show good agreement with each other. The full spectrum approximates a power-law dependence across the entire measured frequency range, though the spectral exponent $\beta$ clearly varies in frequency. These measurements also corroborate the agreement between measurements of charge noise based on coherent spin manipulation on one hand and conductance fluctuations of sensor dots on the other hand. This fact, together with other recent results, including refs. [12,13], suggests that conductance measurements, which are straightforward to implement, can accurately characterize the charge-noise environment. These findings also provide further evidence that the low-frequency noise and the high-frequency noise may have a common physical origin.

Second, our results also provide further evidence for the presence of an inhomogeneous distribution of two-level systems [11,40] as the noise spectrum does not have a uniform exponent. Indeed, deviations in $\beta$ are commonly observed in spectral noise measurements in Si-based devices [11,13,32,41]. These observations emphasize that characterizations of root-mean-square detuning or electrochemical-potential fluctuations do not fully characterize the charge-noise spectrum.

Third, our measurements highlight several differences between Si/SiGe spin qubits and GaAs spin qubits. The high-frequency noise we measure, $S_\mu(1 \text{ MHz}) \sim 10^{-18}$ eV²/Hz, is significantly larger than the $S_\mu(1 \text{ MHz}) \sim 10^{-22}$ eV²/Hz observed in the GaAs S-$T_0$ qubits in refs. [14,15], which had similar levels of low-frequency noise to the Si/SiGe qubit studied here. Moreover, in these GaAs devices, exchange-echo experiments improved the coherence times by approximately two orders of magnitude. In the Si/SiGe S-$T_0$ qubit here, however, the exchange echo only extends the coherence time by a factor of approximately four (Fig. 2d). In this device, we also observe a relatively weak temperature dependence of the high-frequency noise (see Supplementary Note 4), in contrast to the strong temperature dependence of the GaAs S-$T_0$ qubit of ref. [14]. We also observe that the low- and high-frequency noise have a similar temperature dependence.

Although we cannot make general claims about noise in Si devices, we note that the low-frequency noise we measure, along with our observations of a $1/f$-like charge-noise spectrum over a wide frequency band [5], the limited spin-echo coherence-time improvement [38], and a relatively weak temperature dependence [42], are consistent with previous reports in Si devices [12], and suggest a common origin for low- and high-frequency noise in these devices.

Given that the GaAs heterostructures of refs. [14,15] were doped, while the Si/SiGe heterostructure of the present work is undoped, it is surprising that the present device has much larger high-frequency charge noise. This difference may point to the importance of other aspects of the device architecture, including the particular heterostructure or choice of gate dielectric and metal. References [11,43,44] have suggested that charge-noise in semiconductor nanostructures may depend on the details of the device fabrication. Thus, a more comprehensive and methodical study of charge noise and how it depends on device fabrication may shed further light on this problem.

Finally, multiple theoretical predictions have suggested how different mechanisms of electrical noise in semiconductors might couple to spin qubits [45–53]. While our results cannot yet identify a particular source of the charge noise, we note some points of contact between our experiments and theory. First, previous research has suggested that a non-uniform distribution of fluctuators can lead to a $1/f^\beta$-like noise spectrum with $\beta \neq 1$ (refs. [46,54,55]), as we have observed in this and previous work [11]. Second, previous research has suggested that the effects of charge noise should be similar or perhaps larger in Si quantum dots compared to GaAs quantum dots, in agreement with our observations [46]. Prior work has also estimated the effect on exchange couplings of individual charged fluctuators [46]. These predictions are somewhat less than what we observe, possibly suggesting the presence of multiple fluctuators, in agreement with our observations of a relatively smooth noise spectrum. However, we caution that any quantification of the fluctuator density must account for the particular type and orientation of fluctuator, and the exact device geometry [47].

In summary, through a combination of coherent control and electrical transport measurements, we have characterized the charge noise spectrum in a Si/SiGe quantum-dot device from a few microhertz to a few megahertz. The detuning noise inferred from coherent evolution of an S-$T_0$ qubit agrees with the electrochemical potential noise inferred from incoherent transport of a sensor dot. This agreement corroborates the notion that relatively simple conductance measurements may serve as a practical approach to rapidly quantify and compare qubit charge noise. Our strategy for dynamical decoupling, which does not involve external gradient sources or microwave antennas, provides a straightforward way to characterize high-frequency noise without added device complexity. This work also demonstrates the capabilities of Si S-$T_0$ qubits for charge-noise spectroscopy and represents a critical step toward implementing noise-mitigation strategies, such as dynamically-corrected gates, by providing a detailed measurement of the charge-noise spectrum. In light of our findings, future efforts devoted to understanding the impact of fabrication on charge noise seem especially worthwhile. Another important future

avenue of exploration involves measuring charge noise associated with barrier-controlled exchange gates, which are frequently used to implement two-qubit gates between spins. Given the critical importance of suppressing charge noise for realizing the highest-possible gate fidelities in quantum-dot spin qubits, continued progress in understanding and mitigating charge noise is essential.

After completing this manuscript, we became aware of a related result[56].

## Methods

**Device**. The S-$T_0$ qubit device used in this work is fabricated on a Si/SiGe heterostructure with a quantum well of natural Si nominally 50 nm below the surface of the semiconductor. Prior to gate deposition, we deposit 15 nm of $Al_2O_3$ on the surface of the semiconductor via atomic layer deposition. Voltages applied to three layers of overlapping aluminum gates are used to define a double quantum dot, as well as an additional quantum dot to be utilized as a charge sensor. Bias tees are incorporated in the circuits for gates $P_1$, $P_2$, and T to enable nanosecond-scale voltage pulses. Device geometries were designed to incorporate an rf-reflectometry circuit with the sensor dot to allow for microsecond-timescale spin-state readout[21]. The device is cooled in a dilution refrigerator with a base temperature of ~50 mK and a typical electron temperature at or below 100 mK. Additionally, an externally-applied magnetic field, $B_{ext}$, is applied in the plane of the semiconductor.

To extract the lever arm of $P_{1(2)}$, we measure the charge-sensor signal while sweeping the plunger gate over a charge transition corresponding to dot 1(2) at temperatures between 275 and 350 mK, and then fit these acquired signals to the Fermi-Dirac function with $\alpha_{P1(2)}$ as a fit parameter. In this temperature range, tunneling between the dots and the electron reservoirs is thermally broadened. The lever arm of the charge-sensor plunger gate, $\alpha_S$, is extracted from the slopes of Coulomb blockade diamonds.

**Singlet- and random-state initialization**. The qubit state can be initialized as a (4,0) singlet, $|S\rangle$, via electron exchange between dot 1 and its reservoir by pulsing the plunger gates to position L of Fig. 1b for a time $T_L$. When $T_L = 3$–$5$ μs, we estimate that the (4,0) singlet initialization fidelity is >99%. After singlet initialization, the gates are quickly ramped back to the idle position at $\epsilon = 0$ mV before subsequent operations are carried out. We can also intialize a random joint spin state, which is useful for the measurement of the Pauli spin blockade region, by first pulsing into (3,0) to empty dot 2, and then pulsing far into (3,1) to populate all spin states.

**$|X\rangle$ state preparation and mapping**. Upon initializing the system as $|S\rangle$, we can prepare $|X\rangle = \frac{1}{\sqrt{2}}\left(|S\rangle + |T_0\rangle\right)$ in one of two ways: (i) adiabatically separating the electrons by slowly ramping along $\epsilon$ far enough into (3,1) to where $J(\epsilon, V_T) \ll \Delta B_z$[26], or (ii) applying a Hadamard gate ($H$) which takes $|S\rangle$ to $|X\rangle$. The reverse process also maps $|X\rangle$ back to $|S\rangle$. Both methods map $|-X\rangle = \frac{1}{\sqrt{2}}\left(|S\rangle - |T_0\rangle\right)$ to $|T_0\rangle$, and vice versa. Because the minimum value of $J$ is relatively large in our device, compared with $\Delta B_z$, the visibility of exchange oscillations is greater when using the Hadamard gate for preparation and readout rather than adiabatic preparation and readout.

**Readout**. We use rf reflectometry to measure the S-$T_0$ qubit. Applying the rf tone only during the measurement phase ensures that it does not contribute to dephasing. We characterize our readout following the procedures described in ref. [21]. Supplementary Fig. 1a shows a plot of the average (singlet-triplet) single-shot readout fidelity as a function of integration time $t_{avg}$. All data presented in the main text were acquired with integration times between $t_{avg} = 6$–$10$ μs, corresponding to average fidelities greater than 95.6%. A histogram of 10,000 single-shot measurements of randomly-initialized spin states analyzed with an integration time of $t_{avg} = 8.3$ μs is shown in Supplementary Fig. 1b. Supplementary Fig. 1c shows the measured charge sensor signal during triplet-to-singlet decay with a characteristic time of $T_1 = 447$ μs. In Supplementary Fig. 1b, c, the readout is configured such that the lower-voltage signal corresponds to a singlet state, in contrast to the convention used in Fig. 1b.

**Analysis of standard FID measurements**. For FID data shown in Fig. 2a, c, we fit the exchange oscillations measured at each value of $\epsilon$ to a function

$$P_S(t) = P_0 + \left[P_1 \cos(2\pi Jt) + P_2 \sin(2\pi Jt)\right] \\ \times \exp\left[\left(\frac{-(t-t_0)}{T_2^*}\right)^2\right], \tag{2}$$

with $P_0$, $P_1$, $P_2$, $J$, $t_0$, and $T_2^*$ as free parameters. We then additionally fit the extracted values of $J$ as a function of $\epsilon$ to a function $J(\epsilon) = J_0 + J_1 \exp(\epsilon/\epsilon_0)$, from which we extract $dJ/d\epsilon$.

To estimate the underlying charge noise we assume a noise spectrum of the form $S = A_\mu/f$ and use the extracted values of $dJ/d\epsilon$ and $T_2^*$ to calculate[31]

$$A_\mu = \left[2(\pi T_2^*)^2 \ln\left(\frac{t_{meas}}{2\pi T_2^*}\right)\right]^{-1} \left(\frac{dJ}{d\epsilon}\right)^{-2} \alpha_\epsilon^2. \tag{3}$$

Here, we have assumed that the chemical potential of each dot fluctuates independently, and $\alpha_\epsilon = \left(\alpha_{P1}^{-2} + \alpha_{P2}^{-2}\right)^{-1/2}$ is the voltage-to-energy conversion factor for changes in $\epsilon$. We calculate $A_\mu$ for all $\epsilon$ in Fig. 2c in the range 3–5 mV (where the extracted values of $T_2^*$ and $dJ/d\epsilon$ are the most reliable) and determine an average value $A_\mu = 0.42$ μeV$^2$. Finally, from the determined value of $A_\mu$, we estimate a RMS electrochemical potential noise[12]

$$\sigma_\mu = \sqrt{\int_{1/t_{meas}}^{1/t_{rep}} \frac{A_\mu}{f} df} = 2.7\,\mu\text{eV}. \tag{4}$$

Here, $t_{rep} = 24$ μs is the length of each single-shot experiment.

In the determination of $A_\mu$, and therefore also $\sigma_\mu$, we have disregarded effective fluctuations in $V_T$, which is justified when $\left|dJ/dV_T\right|$ is significantly smaller than $\left|dJ/d\epsilon\right|$. To confirm that this criterion is satisfied in our device, we have taken additional data similar to those shown in Fig. 2c, using the same device but in a different dilution refrigerator. For this auxiliary data (Supplementary Fig. 2), we confirm that $\left|dJ/d\epsilon\right|$ is at least five times larger than $\left|dJ/dV_T\right|$ at all measured values of $\epsilon$, and extract values $A_\mu = 1.18$ μeV$^2$ and $\sigma_\mu = 4.4$ μeV. These values agree reasonably well with the values reported in the main text, given that the two data sets were taken in different dilution refrigerators and in slightly different tunings. We therefore conclude that fluctuations in $V_T$ likely do not strongly affect our results. We emphasize that, as pointed out in the main text, extracting $A_\mu$ and $\sigma_\mu$ from FID measurements, which requires assuming a uniform $1/f$ spectrum, generally yields inaccurate results and the values are only estimates of the true noise power.

**Spectral analysis within the filter-function formalism**
*Definition of the power spectrum*. The spectra reported in this work are single-sided. We use the following definition of the two-sided noise spectrum of a signal $X(t)$:

$$S_X'(\omega) = \lim_{T\to\infty} \frac{1}{T}\left|\hat{X}(\omega)\right|^2 \tag{5}$$

where

$$\hat{X}(\omega) = \int_{-T/2}^{T/2} dt X(t) e^{i\omega t}. \tag{6}$$

Using this definition, we also have

$$S_X'(\omega) = \int_{-\infty}^{\infty} G(\tau) e^{i\omega\tau} d\tau, \tag{7}$$

from the Wiener–Khinchin Theorem[57], where $G(\tau) = \langle X(\tau)X(t+\tau)\rangle$ is the auto-correlation function. A single-sided noise spectrum is given by $S_X(\omega) = 2S_X'(\omega)$ such that

$$\int_0^\infty d\omega S_X(\omega) = \int_{-\infty}^\infty d\omega S_X'(\omega). \tag{8}$$

Lastly, we define the normalization of the Dirac-delta function as

$$2\pi\delta(x) = \int_{-\infty}^\infty d\omega e^{i\omega x}. \tag{9}$$

Note with this normalization, we have

$$\begin{aligned} \int_{-\infty}^\infty \frac{d\omega}{2\pi} S_X'(\omega) &= \int_{-\infty}^\infty \frac{d\omega}{2\pi} \int_{-\infty}^\infty d\tau G(\tau) e^{i\omega\tau} \\ &= G(0) \\ &= \sigma_x^2. \end{aligned} \tag{10}$$

*CPMG experiments*. For CPMG experiments in general, including spin-echo experiments, the phase of the qubit, $\phi(t)$, is given by

$$\phi(t) = 2\pi \int_{-\infty}^\infty dt' \nu(t') y(t'; \tau, t_\pi, n_\pi), \tag{11}$$

where $\nu(t')$ is the time-dependent qubit frequency, and $y(t'; \tau, t_\pi, n_\pi)$ is an experiment-specific function that characterizes the pulse sequence[5,31]. As defined in the main text, $\tau$, $t_\pi$, and $n_\pi$ are the total evolution time, the $\pi$-pulse time, and the number of $\pi$ pulses, respectively. In the case of a CPMG experiment, $y(t'; \tau, t_\pi, n_\pi)$ flips between $+1$ and $-1$ with each $\pi$ pulse, and is equal to 0 during all portions of the sequence when the qubit is not freely evolving, including during the finite duration of the $\pi$ pulses[39]. Supplementary Fig. 3 shows an example of $y(t'; \tau, t_\pi, n_\pi)$ for a CPMG experiment with $n_\pi = 8$. By assuming that fluctuations in $\nu(t')$, and therefore $\phi(t)$, are Gaussian-distributed, the average phase accumulation of the

qubit is given by

$$\langle \exp[i\phi(t)] \rangle = \exp\left(-\frac{\langle\phi(t)^2\rangle}{2}\right) = \exp[-\chi(t)], \quad (12)$$

where $\chi(t)$ is the decay envelope. By invoking the relation given in Eq. (10), and defining the spectral weighting function

$$W(f) = |\hat{y}(f; \tau, t_\pi, n_\pi)|^2, \quad (13)$$

where $\hat{y}(f; \tau, t_\pi, n_\pi)$ is the Fourier transform of $y(t; \tau, t_\pi, n_\pi)$, we obtain

$$\begin{aligned}\chi(t) &= \frac{1}{2}\int_{-\infty}^{\infty} df S'_\phi(f) \\ &= 4\pi^2 \int_0^\infty df S'_\nu(f) |\hat{y}(f; \tau, t_\pi, n_\pi)|^2 \\ &= 4\pi^2 \int_0^\infty df S'_\nu(f) W(f; \tau, t_\pi, n_\pi).\end{aligned} \quad (14)$$

Finally, we define the experiment-specific frequency-domain filter function as

$$F(f; \tau, t_\pi, n_\pi) = 4\pi^2 f^2 W(f; \tau, t_\pi, n_\pi). \quad (15)$$

The definition of the filter function of a CPMG experiment with $n_\pi$ refocusing pulses of duration $t_\pi$ and total evolution time $\tau$ is given by Biercuk et al.[39]

$$\begin{aligned}F(f; \tau, t_\pi, n_\pi) = \Big| &1 + (-1)^{n_\pi+1} e^{i2\pi f(\tau + n_\pi t_\pi)} \\ &+ 2\sum_{j=1}^{n_\pi} (-1)^j e^{i2\pi f \delta_j(\tau + n_\pi t_\pi)} \cos(\pi f t_\pi) \Big|^2,\end{aligned} \quad (16)$$

where $\delta_j(\tau + n_\pi t_\pi)$ is the time of the center of the $j^{th}$ $\pi$ pulse. We can rewrite Eq. (14) as

$$\chi(t) = \int_0^\infty df S'_\nu(f) \frac{F(f; \tau, t_\pi, n_\pi)}{f^2}. \quad (17)$$

Note that each CPMG experiment is most sensitive to noise at the frequency corresponding to the maximum value of $W(f; \tau, t_\pi, n_\pi)$, not the maximum value of $F(f; \tau, t_\pi, n_\pi)$.

**Analysis of spin-echo measurements.** At each $\tau$ in a given spin-echo experiment, we extract the echo amplitude from a fit of the singlet return probability to a function of the form

$$\begin{aligned}P_S(\delta t) = P_0 &+ A^e \cos(\omega \delta t - \phi) \\ &\times \exp\left[(-(\delta t - t_0)/T_2^*)^2\right]\end{aligned} \quad (18)$$

(Supplementary Fig. 4a). Here, $P_0$, $A^e$, $\omega$, $\phi$, $t_0$, and $T_2^*$ are all fit parameters. $A^e$ is the amplitude of the oscillations. Supplementary Fig. 4b shows a typical example of the resulting extracted echo amplitude (normalized) as a function of $\tau$ from which we extract information regarding the noise.

For a spin-echo experiment, $n_\pi = 1$, and Eq. (16) simplifies to

$$\begin{aligned}F(f; \tau, t_\pi, n_\pi = 1) = 4\big[&\cos(\pi f t_\pi) \\ &- \cos(\pi f(\tau + t_\pi))\big]^2.\end{aligned} \quad (19)$$

Given this filter function, if the noise spectrum has a form $S(\omega) = A/\omega^\beta$, $\chi(\tau)$ has the form

$$\begin{aligned}\chi(\tau) = -&A2^{-\beta}\pi^{-1}\Gamma(-1-\beta)\sin\left(\frac{\pi\beta}{2}\right)(\tau + t_\pi)^{\beta+1} \\ &\times \big(-|R-1|^{\beta+1} + 2^\beta + 2^\beta R^{\beta+1} - R(R+1)^\beta \\ &- (R+1)^\beta\big),\end{aligned} \quad (20)$$

where $A$ has units of $(\text{rad} - \text{Hz})^{\beta+1}$. For each spin-echo measurement we extract values of $A$, $\beta$, and $T_2^e$ (we define $T_2^e$ as the value of $\tau$ corresponding to $\chi(\tau) = 1$) by fitting the decay curve to a function of the form

$$A_0(\tau) = C \exp[-\chi(\tau)], \quad (21)$$

with $C$, $A$, $T_2^e$, and $\beta$ as fit parameters (Supplementary Fig. 4b). Finally, we convert $A$ to a single-sided charge-noise magnitude, with units of eV$^2$/Hz, via

$$A_\mu = \left(\frac{2A}{(2\pi)^{\beta+2}}\right)\left(\frac{dJ}{d\epsilon}\right)^{-2}\alpha_\epsilon^2. \quad (22)$$

**Analysis of CPMG measurements.** Given the form of $y(t; \tau, t_\pi, n_\pi)$, its mean square value is given by

$$\sigma_y^2 = \frac{tD}{T}, \quad (23)$$

where $D = \tau/t$ is the duty cycle of the sequence, as defined in the main text.

According to Eqs. (5) and (10) we also have

$$\begin{aligned}\int_{-\infty}^{\infty} df S'_y(f) &= \frac{1}{T}\int_{-\infty}^{\infty} df |\hat{y}(f; \tau, t_\pi, n_\pi)|^2 \\ &= \frac{1}{T}\int_{-\infty}^{\infty} df W(f; \tau, t_\pi, n_\pi) \\ &= \sigma_y^2.\end{aligned} \quad (24)$$

Relating Eqs. (23) and (24), we can see that

$$\int_0^\infty df W(f; \tau, t_\pi, n_\pi) = \frac{tD}{2}. \quad (25)$$

For large $n_\pi$ and not-too-small $D$, the spectral weighting function is strongly peaked at odd multiples of $f_0 = n_\pi D/(2\tau)$, with a decreasing contribution from each higher harmonic[5]. Thus, we can approximate $W(f; \tau, t_\pi, n_\pi)$ as

$$W(f; \tau, t_\pi, n_\pi) \approx \frac{Dt}{2}\delta(f - f_0). \quad (26)$$

Finally, using Eqs. (14) and (26) we arrive at

$$\chi(t) = 2\pi^2 Dt S'_\nu(f_0), \quad (27)$$

and therefore

$$S_\mu(f_0) = -\frac{\ln(A^{n_\pi})}{\pi^2 Dt}\left(\frac{dJ}{d\epsilon}\right)^{-2}\alpha_\epsilon^2, \quad (28)$$

which allows us to directly calculate the noise spectral density from the amplitude of the refocused oscillations for individual data points in each CPMG experiment.

In total, and excluding spin-echo experiments, we perform CPMG with $n_\pi = 2$, 3, 4, 6, 8, 10, 12, 14, 16, 24, 32, 48, 64, 96, 128, 256, and 512. For each experiment, we extract the refocused oscillation amplitude, $A^{n_\pi}$, via the same procedure used for spin-echo experiments outlined above. We calculate the noise spectral density according to Eq. (28) for only the subset of CPMG data points corresponding to $n_\pi \geq 8$, $D > 0.2$, and $0.15 < A^{n_\pi} < 0.85$. The first two criteria ensure that $W(f; \tau, t_\pi, n_\pi)$ is strongly peaked at $f = n_\pi D/(2\tau)$. The third constraint is imposed in order to remove those data points which are highly susceptible to noise associated with the experimental setup.

As $D$ decreases, the peak in $W(f; \tau, t_\pi, n_\pi)$ shifts to lower frequencies, and the peaks at higher harmonics tend to increase in strength. Thus, the assumption of Eq. (26) becomes increasingly less accurate. We assess the resulting error for a power-law noise spectrum by assuming a hypothetical noise spectrum $S(f) = A/f^\beta$. We numerically calculate the ratio $\eta = S_{est}(f_0)/S(f_0)$, where $S_{est}(f_0)$ is the estimated spectral density, which is calculated using the approximation of Eq. (26) for the pulse parameters considered in this work. In all cases, $0.5 < \eta < 1$, indicating that the noise values we report in Fig. 5 that correspond to CPMG measurements are likely underestimations of the true value of the noise, but only by a factor of 2 at most.

Supplementary Fig. 5 shows a plot of the $D$ and $n_\pi$ values for each CPMG data point, as well as the expected error associated with the data points used in the calculation of the noise spectrum.

**Analysis of 3-day FID experiment**

*Calculation of the noise spectrum.* We repeatedly measure FID measurements for a total measurement time of 71.81 h. To extract the corresponding noise spectrum, we analyze these measurements according to the following procedure. We first calculate $J(\epsilon)$ by fitting values extracted from a separate FID measurement taken immediately prior to the long FID experiment to $J(\epsilon) = J_0 + J_1 \exp(\epsilon/\epsilon_0)$[14]. We then fit each oscillation trace to a function of the form

$$P_S(t) = A\cos(2\pi Jt + \phi)\exp[-(t/T_2^*)^2] + C, \quad (29)$$

where $A$, $J$, $\phi$, $T_2^*$, and $C$ are fit parameters, to extract the value of $J$. We then convert $J(t_{meas})$ to $\epsilon(t_{meas})$ using $J(\epsilon)$, and finally convert this noisy signal in $\epsilon$ to electrochemical potential noise using $\alpha_\epsilon$.

While each individual column of data in Fig. 3a takes ~0.97 s to acquire, the data acquisition rate is not constant over the entire experiment. During our measurement, we save the data to disk every 1000 repetitions. This periodic saving adds an intermittent delay (Supplementary Fig. 6). Because the effective sampling rate is therefore not constant, we use the Lomb–Scargle method[34], which calculates spectra of unevenly-sampled discrete signals. In our case, because most of the data is acquired at a constant rate, the Lomb–Scargle method produces results nearly identical to a standard periodogram. Lomb–Scargle periodograms are calculated using MATLAB's built in `plomb` function with no oversampling. We also use Bartlett's method to reduce the variance of the estimated periodograms, as outlined in the main text.

Because the FID measurement from which we extract $J(\epsilon)$ only explicitly measures $J(\epsilon)$ in the range $\epsilon = 2$–5 mV, we omit any points in the signal where $\epsilon(t_{meas}) < 2$ mV (only 238 of the 262,144 traces) when calculating the spectrum shown in Fig. 5, as well as during the analysis of noise correlations discussed below. All 262,144 traces are shown in Fig. 3a, b.

We note that this method of calculating the noise spectrum, specifically the function to which the individual traces are fit (Eq. (29)), implicitly relies on the

assumption of $1/f$ noise. To ensure this does assumption does not effect the extracted spectrum, we additionally extract $J(t_{meas})$ from the FFT of the individual traces, and then calculate the power spectrum following the same procedure outlined above. We find that the noise spectrum generated using this method is nearly identical to the spectrum presented in Fig. 5 out from the lowest frequencies out to a few mHz, at which point the noise power falls below the noise floor of the FFT-based method. We thus conclude that the $J(t_{meas})$ signal extracted from fits to the oscillation traces are not appreciably affected by the presence of the Gaussian decay envelope in Eq. (29).

*Noise correlation.* For each column of data shown in Fig. 3a, we generate a histogram of the corresponding single-shot measurements and fit these histograms to equations (1) and (2) of ref. [22]. Among the fit parameters in these equations are the mean sensor signals corresponding to the singlet and triplet outcomes. Assuming that the sensor tuning does not change significantly during the run, changes in the mean singlet and triplet signals are linearly related to changes in the sensor dot electro-chemical potential. Using these fit parameters, we create a time series of the mean singlet signal, $S_{1,s}(t_{meas})$, with one point for each column of Fig. 3a. This time series is generated from the same data used to generate $\epsilon(t_{meas})$ and is thus a concurrent measurement of fluctuations in the sensor dot electrochemical potential. We calculate a Pearson correlation coefficient of $\rho = -0.32$ between $\epsilon(t_{meas})$ and $S_{1,s}(t_{meas})$ for the entire 3-day time series. Supplementary Fig. 7a shows a plot of the normalized signals, along with straight-line fits to each signal, which show that the long-time drift contributes to the anticorrelation. To remove the effect of this slow drift, we divide the signal into 1-h segments (Supplementary Fig. 7b) and calculate an average correlation coefficient across the segments of $\rho = -0.15$ with a standard deviation of $\sigma_\rho = 0.19$. Thus, at timescales between 1 s and 1 h, the double-dot detuning and sensor electrochemical potential are not significantly correlated. This result further justifies the assumption of uncorrelated noise between quantum dots that we have invoked elsewhere. Tracking the mean triplet signal, instead of the mean singlet signal, yields nominally identical results. Additionally, the lack of significant correlations implies that the charge noise is relatively short-wavelength compared to the relevant dot-separation distance of 225 nm (dot 1 to the sensor dot).

**Analysis of charge sensor time series.** We measure the charge-noise spectra of the sensor quantum dot by sampling the reflectometry signal, at a sampling rate $f_s = 100$ kHz for a total time $T = 500$ s when the plunger gate of the dot is set on the side of a transport peak, such that $|dS_1/dV_S|$, the sensitivity of the reflected signal to shifts in the electrochemical potential of the dot, is large. Voltage-noise spectra (in units of $V^2$/Hz) are acquired from the measured signals, $S_1(t)$, via

$$S_{S1}(f) = \frac{2|\tilde{S}_1(f)|^2}{\Delta f N^2},$$ (30)

where $\Delta f = 1/T$ is the frequency interval, $\tilde{S}_1(f)$ is the FFT of $S_1(t)$, and $N = f_s T$ is the total number of points in the signal $S_1(t)$. We convert the voltage noise spectrum to a charge noise power spectrum through

$$S_\mu(f) = \frac{S_{S1}(f)\alpha_S}{|dS_1/dV_S|}.$$ (31)

We extract $dS_1/dV_S$, from a fit of the transport peak shape[58].

We verify that this measurement is sensitive to device noise and establish an effective noise floor of the measurement by repeating the measurement in the Coulomb blockade region where $|dS_1/dV_S| \approx 0$. In this tuning, the acquired noise spectrum is much lower in magnitude and approximately white, verifying that the colored noise we otherwise measure is noise in the device (Supplementary Fig. 8a).

We observe that the charge noise spectra depend on the tuning of the device as well as the amplitude of the rf carrier. Supplementary Fig. 8b shows the effect of the total room-temperature attenuation applied to the carrier on the acquired spectra. In general, more attenuation reduces the charge noise spectrum between 1 Hz and 1 kHz, suggesting that the rf carrier may induce charge fluctuations in this frequency range. All data displayed in the main text are acquired with 40 dB room-temperature attenuation. Supplementary Fig. 8c shows spectra acquired when $\epsilon = 0$mV (inside the Pauli spin blockade region) and when $\epsilon = 30$ mV (near the center of the (3,1) charge region). We hypothesize that the increased charge noise when $\epsilon = 0$ may result from electrons hopping between dots, or between the dots and the reservoirs, both of which are more likely to occur when $\epsilon = 0$. It is likely that these hopping events are induced by the rf carrier, because the excess noise occurs also occurs between 1 Hz and 1 kHz. During S-T$_0$-qubit experiments, the rf carrier is unblanked only during readout.

## Data availability
The processed data are available at https://doi.org/10.5281/zenodo.5874151[59]. The raw data are available from the corresponding author upon reasonable request.

## Code availability
The code used to analyze the data in this work is available from the corresponding author upon reasonable request.

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

## Acknowledgements
Research was sponsored by the Army Research Office and was accomplished under Grant Numbers W911NF-16-1-0260 and W911NF-19-1-0167. The views and conclusions contained in this document are those of the authors and should not be interpreted as representing the official policies, either expressed or implied, of the Army Research Office or the U.S. Government. The U.S. Government is authorized to reproduce and distribute reprints for Government purposes notwithstanding any copyright notation herein. E.J.C. was supported by ARO and LPS through the QuaCGR Fellowship Program.

## Author contributions
E.J.C., J.N., and J.M.N. conceptualized the experiment, and conducted the investigation. L.F.E. provided resources. E.J.C. and J.M.N. wrote the manuscript. J.M.N. supervised the effort.

## Competing interests
The authors declare no competing interests.
