## [Peer Review File · Nature Communications]

REVIEWER COMMENTS

Reviewer #1 (Remarks to the Author):

The Authors report on the charge noise of a silicon double quantum dot over 13 decades. Despite I believe that the results are correct and the description of the experiments adequate for reproducibility of the method, the article lacks of generality and of interest for a broad Readership.

It looks more a narration of an extensive research with multiple techniques of interest for the Readership of a specialized measurements journal like Rev. Sci. Instr. or some IEEE Transaction, than a milestone scientists should be aware of. The Authors look to agree with my conclusion: they also start their conclusion by declaring "In summary, we have demonstrated coherent control of a singlet-triplet qubit in a Si/SiGe heterostructure with an overlapping-gate architecture." but this kind of experiment is done in many research labs and it has been published several times in the last ten years. The extended - multi-method - noise characterization in their special situation is not sufficient to make the article suitable for Nature Communication. I recommend publication on a specialized journal.

Reviewer #2 (Remarks to the Author):

The study of charge noise is one of the most pressing issues in solid state quantum computing, and is a topic where considerable theory exists but not enough experimental measurements to confirm or deny theoretical predictions. Part of the reason for this is that, using the system at hand as an example, background charge fluctuations affect quantum dots in a particular way, causing fluctuations in their energy levels, as well as, more importantly, between their energy levels. Therefore, despite the wealth of noise studies in devices from the 1980s none of their findings are applicable to solid state qubits, and what is needed is studies of noise effects at the location of the qubit itself. In this general context the present paper presents a very important study, which is timely in the field. The study is thorough and the conclusions revealing, in particular regarding the relative size of the effect in Si vs GaAs, and the agreement between conductance and coherence measurements. The paper could be considered for publication, but before making a decision I have the following questions:

1. The authors state the orbital splitting rather than the valley splitting is the limiting factor for the orbital configuration studied here. What is the rationale behind this statement?

2. The authors state that dJ/dV_T is negligible. Why is this the case for the setup used here? Normally one expects dJ/dE to make the biggest contribution, as the authors assume, but this assumption is frequently made for reasons of convenience. Do the authors have solid grounds for not considering dJ/dV_T ?
3. The results and trends for T_2^* should be compared to relevant theoretical predictions in the literature, for example Appl. Phys. Lett. 95, 073102 (2009).
4. Is there any possibility of extracting the temperature dependence of A_{μ} beyond what is presented in S4b? This temperature dependence is usually assumed to be linear, and it would be interesting to investigate whether this is justified for qubits.

Reviewer #3 (Remarks to the Author):

The manuscript by Connors et al describes a comprehensive series of experiments that examine charge noise in Si/SiGe spin qubit devices over several orders of magnitude in frequency by combining several measurement techniques, ranging from direct charge sensing to noise inferred from dynamical decoupling measurements of the singlet-triplet qubit. The primary claim is that for these devices, the power spectral density of the noise remains close to, but not exactly, $1/f$ over 13 decades in frequency and that basic transport measurements can be used to characterize the charge noise environment in spin qubit devices rather than relying on more complex coherent measurements. The secondary claim is that the deviations from a pure $1/f$ spectrum mean that “root-mean-square” noise measurements are an incomplete measure of the noise, and while I agree with the claim I am not sure it’s that novel or important. Also, such a claim would be best supported by measurements from multiple devices rather than a single one. The last claim compares these results to similar experiments with GaAs spin qubits where, for GaAs, the high frequency part of the noise spectrum was significantly below a simple extension of the low-frequency $1/f$ noise. This is perhaps less of a claim and more of an observation that is interesting, but is certainly supported by the measurements.

These claims, at least the first, are novel and of interest to people working with Si/SiGe qubits as well as those in the more general solid-state quantum device community. Overall, I find the work to be valid and convincing, and generally in support of the primary claims. That being said, I do have several comments or points that should be addressed prior to publication.

1. The introduction seems to imply that the noise on the gate voltages themselves are a major cause of error in spin qubit devices. I think the authors are trying to say that qubit operation depends on the electric potential, which is composed of a potential applied by the gate voltages plus any other

charge fluctuations in the environment. I would suggest to re-phrase the sentence to clarify especially for readers of the broader community.

2. The authors contrast their approach with others including reference 5, where a micromagnet is used to create a magnetic field gradient and say “The approach we present here requires two electrons in a double quantum dot, but does not require the generation of magnetic fields or gradients...”. However, a key component of their scheme is that an effective out-of-plane magnetic field gradient is created through the application of an in-plane magnetic field and the recently discovered (Refs 12,13 of the supplementary) g -factor anisotropy between dots resulting from spin-orbit coupling. Doesn’t that directly undermine the claim? They still needed to apply a magnetic field in order to have the required 2 interactions to control the qubit state. I understand that the device design is simpler because you don’t need a micromagnet, but you do need a magnetic field and you do need the g -factor effect for it to work, which both reduces the validity of the claim and limits the application of these methods to Si/SiGe qubits. My recommendation would be to soften the remark to make it more consistent with what follows.

3. The authors do not specify if these devices were fabricated using isotopically enriched materials or not. I think this is an important aspect of the approach because time-dependent magnetic gradients due to non spin-0 nuclei would result in miscalibrations of the H-gate over timescales related to nuclear spin dephasing. The authors should clarify and state the impact on their devices as the approach may not be possible using devices without enriched materials, and would prevent reproduction of the results.

4. The authors state that in the 3-electron charge configuration, the two lowest energy electrons form a singlet and occupy the ground valley state. The remaining electron acts like a “valence” electron and lives in the excited valley state. This isn’t exactly true and depends on how the noise environment (electron temperature) compares to the valley energy. Presumably, the electrons occupy a thermal mixture of ground valley and excited valley states. I don’t think this changes the result, but rather it is important to emphasize that in this multi-electron operation mode the valley splitting still matters and such a simple interpretation is not necessarily true.

5. The ST Hamiltonian should be presented as an “effective Hamiltonian” because it is a bit confusing to the general audience as S_z and S_x are usually thought of as the standard Pauli operators on a spin-1/2 particle. The Hamiltonian shown is an effective Hamiltonian in the singlet-triplet subspace when mapped onto a spin-1/2 particle. This is certainly a standard interpretation for ST qubit specialists, but perhaps isn’t obvious to the non-specialized reader.

6. I think the authors need to be explicit in the main text that their longitudinal Zeeman splitting results from the physics described in References 12,13 of the supplementary. By only pointing to the SM, it's very confusing as to how this term comes about and it's critical for understanding how the rest of the story unfolds. It also makes it challenging to understand how widely applicable this technique (forming the Hadamard out of a combination of exchange and Zeeman splitting) is to other systems.

7. The authors state that when $V_T = 0$, they are unable to achieve the standard operating point of $dB_z \gg J$, which is how they end up using this Hadamard gate. However, why is $V_T=0$ the limit? Can they not apply a reverse bias to further reduce the tunnel coupling between dots? One of the purported advantages of S-T qubits is that the exchange interaction can be varied over orders of magnitude by relatively small adjustments in gate voltage, but that doesn't appear to be the case in this device. Is this something that the authors can address? As of now, it reads a bit like "making lemonade out of lemons" and reduces the impact and applicability of the work.

8. The authors neglect effective fluctuations in V_T for their analysis, which is largely used to make a legitimate comparison between the charge noise measured through coherent measurements (using detuning operation) to more basic transport measurements. However, the community is focused on using "symmetric exchange" for actual qubit gate operation because it reduces the impact of charge noise. Can the authors say anything about what limits performance in this mode of operation (as opposed to detuning) because it really is the symmetric mode that matters for qubit performance? This would significantly increase the impact of the work if it could be extended in this manner. Is it possible to extend the analysis to the place where $dJ/d\epsilon = 0$?

9. What limits the contrast of the oscillations in Figure 2b? Is this an initialization issue or a gate calibration issue?

10. In the analysis which extracts charge noise from the measured T_2^* and $dJ/d\epsilon$ values, do the authors assume that the chemical potential on each dot fluctuates independently? It seems like their math makes this assumption (expression S3) but they should be explicit that they are doing so. Are there other measurements which validate this assumption? Presumably as the electrons get closer and closer together by moving towards $\epsilon=0$ or by increasing the tunnel coupling this assumption must break down. Has this been studied?

11. The authors could extend their charge sensor measurements to even lower frequency by tracking the location of the Coulomb blockade peak over a long timescale rather than Fourier transforming the RF reflected signal while sitting on the slope of the blockade peak for charge sensing. In principle, this kind of measurement could be performed contemporaneously with the low

frequency FID measurements, which could elucidate if the charge sensor fluctuations are correlated with those of the electrons undergoing coherent FID oscillations. Such a measurement would provide further justification of uncorrelated noise assumptions underpinning the analysis.

12. The authors mention that they observe a weak temperature dependence of the high frequency noise, but what about at low frequency (say through FID or CS measurements)? What does this say about potential sources of the noise?

13. In the supplementary materials, the authors state an initialization fidelity that exceeds the measurement fidelity by a considerable margin. How is that possible? Further, is this initialization fidelity the probability of forming a (4,0) singlet or a (3,1) singlet?

14. How sensitive is the optimal Hadamard gate to the control parameters? It might be nice to see a plot of the optimization landscape to give a sense for how well this must be controlled. How often does it need to be recalibrated?

To the Reviewers:

Thank you for your thorough and constructive comments. We have addressed each of the comments below. The comments from the reviewers are reproduced in black text, and our responses appear in blue text. At the end of this document, we append a complete list of changes to the manuscript. The manuscript itself is marked up to show the changes we have made.

Sincerely, on behalf of the authors,

John Nichol

Reviewer #1 (Remarks to the Author):

The Authors report on the charge noise of a silicon double quantum dot over 13 decades. Despite I believe that the results are correct and the description of the experiments adequate for reproducibility of the method, the article lacks of generality and of interest for a broad Readership. It looks more a narration of an extensive research with multiple techniques of interest for the Readership of a specialized measurements journal like Rev. Sci. Instr. or some IEEE Transaction, than a milestone scientists should be aware of. The Authors look to agree with my conclusion: they also start their conclusion by declaring "In summary, we have demonstrated coherent control of a singlet-triplet qubit in a Si/SiGe heterostructure with an overlapping-gate architecture." but this kind of experiment is done in many research labs and it has been published several times in the last ten years. The extended - multi-method - noise characterization in their special situation is not sufficient to make the article suitable for Nature Communication. I recommend publication on a specialized journal.

We thank the reviewer for the positive appraisal of the accuracy and description of our experiments. We also thank the reviewer for bringing to our attention that we had not provided enough context for our work. We have added text to the introduction and conclusion emphasizing the facts that charge noise is one of the most pressing difficulties facing spin qubits, but there are not enough experiments that show how to mitigate it, or that confirm or deny theoretical predictions surrounding it. In this context, our experiments and measurements are an important step forward for Si spin qubits.

Reviewer #2 (Remarks to the Author):

The study of charge noise is one of the most pressing issues in solid state quantum computing, and is a topic where considerable theory exists but not enough experimental measurements to confirm or deny theoretical predictions. Part of the reason for this is that, using the system at hand as an example, background charge fluctuations affect quantum dots in a particular way, causing fluctuations in their energy levels, as well as, more importantly, between their energy levels. Therefore, despite the wealth of noise studies in devices from the 1980s none of their findings are applicable to solid state qubits, and what is needed is studies of noise effects at the location of the qubit itself. In this general context the present paper presents a very important study, which is timely in the field. The study is thorough and the conclusions revealing, in particular regarding the relative size of the effect in Si vs GaAs, and the agreement between

conductance and coherence measurements.

The paper could be considered for publication, but before making a decision I have the following questions:

We thank the reviewer for the positive recommendation of the paper.

1. The authors state the orbital splitting rather than the valley splitting is the limiting factor for the orbital configuration studied here. What is the rationale behind this statement?

In the (3,1) charge configuration, the lowest valley level (k) of the left dot is full. The excited valley (k') of the left dot has a single electron in it. When the “valence” electrons in the left and right dots have the singlet configuration, the electron in the right dot can tunnel into the k' valley. However, when the valence electrons have the triplet configuration, the electron in the right dot cannot tunnel into the k' state of the left dot, due to the Pauli exclusion principle. Instead, the lowest available state for it to tunnel into is the ground state valley associated with the first excited orbital state. This approach effectively enables a large singlet-triplet splitting in the presence of a small valley splitting and has been used in other works (e.g. Ref [22]) for this purpose.

2. The authors state that dJ/dV_T is negligible. Why is this the case for the setup used here? Normally one expects dJ/dE to make the biggest contribution, as the authors assume, but this assumption is frequently made for reasons of convenience. Do the authors have solid grounds for not considering dJ/dV_T ?

All noise spectroscopy measurements are made by operating the qubit either entirely or primarily via detuning control as opposed to barrier control. In this regime, $|dJ/d\epsilon|$ is at least 6.5 x larger than $|dJ/dV_T|$. To confirm that fluctuations in V_T do not affect our analysis of FID oscillations (see Fig. 2c and the discussion on page 3 in the main text), we have taken an additional data set and analyzed the data for ϵ values where $dJ/d\epsilon$ is at least 5 times larger than dJ/dV_T , and we find comparable results to what we had reported earlier for A_μ and σ_μ , confirming that fluctuations in V_T do not substantially affect our results. This additional dataset is discussed in the Supplemental Material. We note that this dataset was taken with the same device but in a different dilution refrigerator.

3. The results and trends for T_2^* should be compared to relevant theoretical predictions in the literature, for example Appl. Phys. Lett. 95, 073102 (2009).

We thank the reviewer for pointing out that we neglected to discuss how our work compares to previous theoretical results. We have added a paragraph to the discussion section to remedy this deficiency. Additionally, we note that the analysis methods we used for the FID, echo, and CPMG experiments are based on the theory discussed in Culcer et al., Appl. Phys. Lett. 95, 073102 (2009) and Cywinski et al, Phys. Rev. B 77, 174509 (2008). Thus, our results are consistent with these works.

4. Is there any possibility of extracting the temperature dependence of A_μ beyond what is presented in S4b? This temperature dependence is usually assumed to be linear, and it would be interesting to investigate whether this is justified for qubits.

We suppose the reviewer means figure S6b rather than S4b. We are not able to measure at temperatures higher than we have shown, because the initialization and/or readout processes of the S-T0 qubit degrade significantly at these temperatures. However, previous work by us (Ref. [11]) and others (Ref. [10]) has confirmed that the noise power does *not* always scale linearly with temperature.

Reviewer #3 (Remarks to the Author):

The manuscript by Connors et al describes a comprehensive series of experiments that examine charge noise in Si/SiGe spin qubit devices over several orders of magnitude in frequency by combining several measurement techniques, ranging from direct charge sensing to noise inferred from dynamical decoupling measurements of the singlet-triplet qubit. The primary claim is that for these devices, the power spectral density of the noise remains close to, but not exactly, $1/f$ over 13 decades in frequency and that basic transport measurements can be used to characterize the charge noise environment in spin qubit devices rather than relying on more complex coherent measurements. The secondary claim is that the deviations from a pure $1/f$ spectrum mean that “root-mean-square” noise measurements are an incomplete measure of the noise, and while I agree with the claim I am not sure it’s that novel or important. Also, such a claim would be best supported by measurements from multiple devices rather than a single one. The last claim compares these results to similar experiments with GaAs spin qubits where, for GaAs, the high frequency part of the noise spectrum was significantly below a simple extension of the low-frequency $1/f$ noise. This is perhaps less of a claim and more of an observation that is interesting, but is certainly supported by the measurements. These claims, at least the first, are novel and of interest to people working with Si/SiGe qubits as well as those in the more general solid-state quantum device community. Overall, I find the work to be valid and convincing, and generally in support of the primary claims. That being said, I do have several comments or points that should be addressed prior to publication.

We thank the reviewer for the positive evaluation of this work.

1. The introduction seems to imply that the noise on the gate voltages themselves are a major cause of error in spin qubit devices. I think the authors are trying to say that qubit operation depends on the electric potential, which is composed of a potential applied by the gate voltages plus any other charge fluctuations in the environment. I would suggest to re-phrase the sentence to clarify especially for readers of the broader community.

We thank the reviewer for pointing out this issue. We have fixed it in the first paragraph.

2. The authors contrast their approach with others including reference 5, where a micromagnet is used to create a magnetic field gradient and say “The approach we present here requires two electrons in a double quantum dot, but does not require the generation of magnetic fields or gradients...”. However, a key component of their scheme is that an effective out-of-plane magnetic field gradient is created through the application of an in-plane magnetic field and the recently discovered (Refs 12,13 of the supplementary) g-factor anisotropy between dots resulting from spin-orbit coupling. Doesn’t that directly undermine the claim? They still needed to apply a magnetic field in order to have the required 2 interactions to control the qubit state. I understand that the device design is simpler because you don’t need a micromagnet, but you do need a magnetic field and you do need the g-factor effect for it to work, which both reduces the

validity of the claim and limits the application of these methods to Si/SiGe qubits. My recommendation would be to soften the remark to make it more consistent with what follows.

The reviewer is correct, although the effective gradient we are sensitive to is an in-plane gradient. We have fixed the wording to emphasize that no additional device-level entities, like a micromagnet or antenna are required for this approach, although we do rely on the uniform external field and g-factor difference.

3. The authors do not specify if these devices were fabricated using isotopically enriched materials or not. I think this is an important aspect of the approach because time-dependent magnetic gradients due to non spin-0 nuclei would result in miscalibrations of the H-gate over timescales related to nuclear spin dephasing. The authors should clarify and state the impact on their devices as the approach may not be possible using devices without enriched materials, and would prevent reproduction of the results.

We have changed the first sentence in the third paragraph to indicate that we are not using isotopically enriched materials. We have also added text later on to the discussion of the H gate calibration in the supplement to indicate that the residual ^{29}Si can potentially introduce errors.

4. The authors state that in the 3-electron charge configuration, the two lowest energy electrons form a singlet and occupy the ground valley state. The remaining electron acts like a “valence” electron and lives in the excited valley state. This isn’t exactly true and depends on how the noise environment (electron temperature) compares to the valley energy. Presumably, the electrons occupy a thermal mixture of ground valley and excited valley states. I don’t think this changes the result, but rather it is important to emphasize that in this multi-electron operation mode the valley splitting still matters and such a simple interpretation is not necessarily true.

We have changed the sentence in question to emphasize that our description is only true for the ground-state configuration of the electrons.

5. The ST Hamiltonian should be presented as an “effective Hamiltonian” because it is a bit confusing to the general audience as S_z and S_x are usually thought of as the standard Pauli operators on a spin-1/2 particle. The Hamiltonian shown is an effective Hamiltonian in the singlet-triplet subspace when mapped onto a spin-1/2 particle. This is certainly a standard interpretation for ST qubit specialists, but perhaps isn’t obvious to the non-specialized reader.

We changed the wording to emphasize that the Hamiltonian is indeed an effective Hamiltonian, and that it refers to the singlet-triplet basis.

6. I think the authors need to be explicit in the main text that their longitudinal Zeeman splitting results from the physics described in References 12,13 of the supplementary. By only pointing to the SM, it’s very confusing as to how this term comes about and it’s critical for understanding how the rest of the story unfolds. It also makes it challenging to understand how widely applicable this technique (forming the Hadamard out of a combination of exchange and Zeeman splitting) is to other systems.

We have addressed this in our response to point 2 above.

7. The authors state that when $V_T = 0$, they are unable to achieve the standard operating point of $dB_z \gg J$, which is how they end up using this Hadamard gate. However, why is $V_T=0$ the limit? Can they not apply a reverse bias to further reduce the tunnel coupling between dots? One of the purported advantages of S-T qubits is that the exchange interaction can be varied over orders of magnitude by relatively small adjustments in gate voltage, but that doesn't appear to be the case in this device. Is this something that the authors can address? As of now, it reads a bit like "making lemonade out of lemons" and reduces the impact and applicability of the work.

The reviewer is correct that, in principle, applying a negative voltage to the barrier gate should suppress residual exchange. We did try this, but we found that it did not work as well as the H gate. One possible reason is that such an X gate would require large, simultaneous changes to both the detuning and barrier gate voltages. We generally find that large, sudden voltage pulses applied to multiple gates are difficult to implement with precision. However, we do not have a complete understanding of this. We also emphasize that the composite $X=HZH$ gate approach we implement is potentially applicable to other systems with restrictions on Hamiltonian parameters.

8. The authors neglect effective fluctuations in V_T for their analysis, which is largely used to make a legitimate comparison between the charge noise measured through coherent measurements (using detuning operation) to more basic transport measurements. However, the community is focused on using "symmetric exchange" for actual qubit gate operation because it reduces the impact of charge noise. Can the authors say anything about what limits performance in this mode of operation (as opposed to detuning) because it really is the symmetric mode that matters for qubit performance? This would significantly increase the impact of the work if it could be extended in this manner. Is it possible to extend the analysis to the place where $dJ/d\epsilon = 0$?

We thank the reviewer for bringing up these points. We agree that barrier-controlled exchange is more popular than detuning-controlled exchange for two-qubit gates between spins, and for Z gates in S-T0 qubits. However, detuning-controlled exchange gates are still needed, for example, for capacitively coupling two S-T0 qubits. Moreover, such an entangling operation is expected to benefit significantly from dynamical decoupling, as we have explored in this work.

However, we agree that studying the noise spectrum associated with barrier-controlled exchange will be essential, and we think it should be the subject of future work. As the reviewer notes, our goals in this work were to connect the noise measured via exchange measurements to noise measured via transport measurements and to compare with other reports in the literature. As is typical for noise measurements in spin qubits, we report chemical potential noise. Extracting chemical-potential noise from barrier-controlled exchange measurements is not easily possible, and effective gate-voltage noise values are usually reported. We have added text to the conclusion to emphasize the importance of noise spectroscopy of barrier-controlled exchange oscillations.

9. What limits the contrast of the oscillations in Figure 2b? Is this an initialization issue or a gate calibration issue?

We suspect that gate calibration and pulse errors diminish the contrast of the oscillations. We suspect, for example, that unintentional adiabaticity in our pulses would favor the singlet, which is the ground state.

10. In the analysis which extracts charge noise from the measured $T2^*$ and $dJ/d\epsilon$ values, do the authors assume that the chemical potential on each dot fluctuates independently? It seems like their math makes this assumption (expression S3) but they should be explicit that they are doing so. Are there other measurements which validate this assumption? Presumably as the electrons get closer and closer together by moving towards $\epsilon=0$ or by increasing the tunnel coupling this assumption must break down. Has this been studied?

We have indeed assumed that the chemical potential of each dot fluctuates independently. We have now clarified this explicitly in the text. We are not aware of any studies of the noise correlation between dots as a function of ϵ . However, the electrons are substantially separated in our work, since the exchange couplings we investigate are significantly less than the singlet-triplet splitting associated with the individual dots.

Inspired by this comment and the one below, we have also analyzed the correlation between the double-dot detuning and the sensor dot chemical potential, and we describe these results in the supplement. These measurements indicate that the noise between the double dot and sensor are only weakly anticorrelated. We now mention this additional analysis in the text.

We also agree with the reviewer's implication that a careful study of the noise correlations between dots would yield extremely useful information relating to the characteristics of the fluctuators.

11. The authors could extend their charge sensor measurements to even lower frequency by tracking the location of the Coulomb blockade peak over a long timescale rather than Fourier transforming the RF reflected signal while sitting on the slope of the blockade peak for charge sensing. In principle, this kind of measurement could be performed contemporaneously with the low frequency FID measurements, which could elucidate if the charge sensor fluctuations are correlated with those of the electrons undergoing coherent FID oscillations. Such a measurement would provide further justification of uncorrelated noise assumptions underpinning the analysis.

On the suggestion of the reviewer, we performed this analysis, using saved data already associated with the three-day FID measurement data. Over the entire 3-day period, we calculate a Pearson correlation coefficient of -0.33 between the double-dot detuning and sensor-dot electrochemical potential, suggesting weak anticorrelations between the sensor dot and the double dot. Taking the average correlation coefficient over one-hour intervals to eliminate drift, we find an average correlation of -0.15, further supporting the assumption of only weak anticorrelations. We have added a description of this analysis, together with additional figures, to the Supplemental Material.

12. The authors mention that they observe a weak temperature dependence of the high frequency noise, but what about at low frequency (say through FID or CS measurements)? What does this say about potential sources of the noise?

The low-frequency and high-frequency noise show similar temperature dependences (see Figures S6b and S6d of the initial submission, which are now Figures S8b and S8d). These low-frequency data are acquired via FID measurements. At the level of this analysis, it suggests that the low- and high-frequency noise have the same origin.

Additionally, a much more thorough study of the temperature dependence of the low-frequency noise in similar devices is given in our previous work [Connors et al. PRB (2019)].

13. In the supplementary materials, the authors state an initialization fidelity that exceeds the measurement fidelity by a considerable margin. How is that possible? Further, is this initialization fidelity the probability of forming a (4,0) singlet or a (3,1) singlet?

The single-qubit singlet-return values we quote are corrected for the readout visibility, so the single-qubit state-vector norms can exceed the readout fidelity. The quoted fidelity is the (4,0) singlet initialization fidelity, and we have modified the supplemental material to clarify this.

14. How sensitive is the optimal Hadamard gate to the control parameters? It might be nice to see a plot of the optimization landscape to give a sense for how well this must be controlled. How often does it need to be recalibrated?

The Hadamard gate is sensitive to the choice of epsilon and gate time. We prefer to calibrate the gate immediately before every echo and CPMG measurement, though in practice, the gate only needs minor recalibration approximately once per day to work reasonably well. As discussed above, errors in the Hadamard gate could likely be minimized in isotopically pure material. We have added additional information regarding frequency of recalibration and included an additional figure describing the calibration procedure, which is automated, to the supplemental material.

Complete list of changes to the manuscript

Main Text Changes:

1. We added Lisa F. Edge of HRL Laboratories, LLC to the author list.
2. We have modified the introduction and conclusion to emphasize the importance of charge noise for Si spin qubits.
3. We have noted that our experiment is carried out in a device with a natural (non isotopically-purified) Si quantum well.
4. We have specified that the four electron configuration we discuss is the ground state, and therefore do not imply that other states are not relevant.
5. We have specified that the Hamiltonian we write down is an effective Hamiltonian for the singlet-triplet qubit in the S-T0 basis.
6. We have specified that we believe dBz to be determined by g-factor differences, and we have mentioned this earlier in the work. We have also added citations in regard to this.
7. We have specified that S_x and S_z are spin $\frac{1}{2}$ operators in the S-T0 basis.
8. We have explicitly noted that the extraction of A_μ and σ_μ hinge on the assumption that neighboring dots fluctuate independently, which is supported by correlation measurements in the Supplemental Material (see below).

9. We have updated Figure 5 to calculate the “FID” portion of the spectrum only from non-clipped data points in the measurement (see below). This resulted in negligible changes to the data shown.
10. We have explicitly noted that the low- and high-frequency noise have similar temperature dependencies.
11. We have noted that our results suggest a common origin of low- and high-frequency noise.
12. We have added a paragraph to the discussion connecting our results to a number of theoretical predictions.
13. We have made minor adjustments regarding typos and formatting.

Supplemental Material Changes:

1. We have added Lisa F. Edge of HRL Laboratories, LLC to the author list.
2. We have noted that our experiment is carried out in a device with a natural (non-isotopically-purified) Si quantum well.
3. We have specified that our quoted initialization fidelity is a (4,0) singlet initialization fidelity.
4. We have provided further details regarding the Hadamard gate calibration, including how often we recalibrate. We have also included an additional figure detailing the calibration procedure.
5. We have explicitly noted that we assume the chemical potential of each dot fluctuates independently.
6. We added a paragraph to the FID measurement analysis section discussing our motivation in disregarding effective fluctuations in the tunneling gate voltage.
7. We have added a short paragraph to the long-FID measurement section detailing that some of the determined values of epsilon were calculated to be outside of the measured range of J vs epsilon. In our initial submission, we calculated these positions by extrapolating the fit of J vs epsilon to these values of epsilon, but we have decided that it is more accurate to omit these data entirely, since we are already using the Lomb-Scargle method of calculating the periodogram. This amounted to discarding only 238 of the total 262144 points in the epsilon vs lab time signal from which we calculate the noise spectrum shown in Figure 5 of the main text. We have updated Figure 5 of the main text to reflect this change.
8. We have added a section discussing the noise correlation between the noise of the sensor quantum dot electrochemical potential, and the detuning noise of the double-dot. We have included a plot of these signals.
9. We have made minor adjustments regarding typos and formatting.

REVIEWER COMMENTS

Reviewer #2 (Remarks to the Author):

The authors have addressed all my points satisfactorily. I recommend the manuscript for publication.

Reviewer #4 (Remarks to the Author):

In this work the authors set out on a broad study of charge noise in Si/SiGe quantum dot devices. Through a variety of means, they probe the charge noise spectrum over 13 orders of magnitude in frequency and find a roughly consistent $1/f$ spectrum throughout, with an amplitude near $0.4 \mu\text{eV}^2/\text{Hz}$. I congratulate the authors on their well-written manuscript describing careful work. Given the immense promise of SiGe devices to the future of quantum computing, I believe this work is of broad enough interest for publication in Nature Communications. With satisfactory responses to the following concerns I will happily recommend this work for publication.

- One element of analysis caused my ears to perk up, though it may be fine. In the FID data analysis, if I understand correctly, the decay curves were fit to sinusoids with a Gaussian envelope, in some sense assuming a $1/f$ spectrum (at relatively higher frequencies). The extracted J_s are then analyzed to assert a $1/f$ spectrum (at relatively lower frequencies). Does the initial assumption of spectrum bleed through to any effect on the extracted spectrum?

- As natural Si is used, hyperfine dephasing is presumably significant, and is also known to have a $1/f$ spectrum. Can you comment on why magnetic dephasing is disregarded in this work?

- Can you comment on the effect of finite-duration pulses in your echo sequences, and any effect may have on exposure to various noise sources?

- Figure 4b seems to show a large spread in estimated spectral exponent as a function of J . What causes this, and is it meaningful?

- In the caption for 4d, it is stated "data points that fall between the horizontal gray lines are used to calculate the underlying noise spectrum." Why isn't all of the data used in this fit?

- Please comment on expected electrical noise. e.g. from control electronics and Johnson noise.

Congratulations again on this excellent work,

Jacob Blumoff

To the Reviewers:

Thank you for your thorough and constructive comments. We have addressed each of the comments below. The comments from the reviewers are reproduced in black text, and our responses appear in blue text. At the end of this document, we append a complete list of changes to the manuscript. The manuscript itself is marked up to show the changes we have made.

Sincerely, on behalf of the authors,

John Nichol

Reviewer #2:

The authors have addressed all my points satisfactorily. I recommend the manuscript for publication.

We thank the reviewer for the recommendation.

Reviewer #4:

- One element of analysis caused my ears to perk up, though it may be fine. In the FID data analysis, if I understand correctly, the decay curves were fit to sinusoids with a Gaussian envelope, in some sense assuming a $1/f$ spectrum (at relatively higher frequencies). The extracted Js are then analyzed to assert a $1/f$ spectrum (at relatively lower frequencies). Does the initial assumption of spectrum bleed through to any effect on the extracted spectrum?

The assumption of a Gaussian envelope does not have a significant impact on the extracted spectrum, because the spectrum only relies on the frequency of the extracted oscillations, not their decay time or envelope.

To verify this claim, we have reanalyzed this data in two additional ways. First, we extract the frequency of each time series by finding the maximum value in the FFT of that trace. Second, we fit the data as before, but we allow the decay exponent to vary between fits. The figure below compares the extracted charge noise power spectra from these two methods, and the original method, which assumes a Gaussian decay. Both fitting methods yield approximately the same power spectrum. The method using the FFT also agrees, though it has a higher noise floor.

We have added a few sentences to the relevant section of the Supplemental Material describing that the implicit assumption of Gaussian decay does not substantially impact the spectrum.

- As natural Si is used, hyperfine dephasing is presumably significant, and is also known to have a $1/f$ spectrum. Can you comment on why magnetic dephasing is disregarded in this work?

In all qubit-dephasing-based measurements of the charge noise (FID, echo, and CPMG), the Hamiltonian governing the evolution of the ST_0 qubit is dominated by the exchange coupling, which is largely insensitive to magnetic field fluctuations (either global or local), but it is highly sensitive to electrical noise via its dependence on the detuning. The inverse relationship of T_2^* on $|dJ/d\epsilon|$ (Fig 2c-d) is a nice indicator that indeed dephasing in this regime is dominated by charge noise. We also assume that small magnetic field changes have negligible effect on the sensor quantum dot and thus do not impact the measurements of charge noise based on rf reflectometry.

- Can you comment on the effect of finite-duration pulses in your echo sequences, and any effect may have on exposure to various noise sources?

We thank the reviewer for bringing this to our attention. In fact, we had not previously considered this in our analysis. The pi pulses in our experiment are rather long (~ 250 ns), so it is indeed important to take this into account. We have redone the analysis of the echo and CPMG experiments to take the duration of the pi pulses. The results of this analysis, which we describe briefly here and in detail in the paper, are consistent with our previous claims, although the bandwidth of our measurements span nearly 12 orders of magnitude, instead of 13 as we had claimed previously.

The primary effect of finite-duration refocusing pulses is to modify the filter function of the pulse sequence [Biercuk et al., PRA **79**, 062324 (2009)]. We have recomputed the theoretical form of the spin-echo decay as described in the supplement, and we have refit the echo decay curves, taking into account the known pi-pulse time, to extract new decay exponents and spectral magnitudes. We have updated Figure 4b and 4c accordingly. We have added a plot to Figure 4c showing the ratio of the pi pulse time to the echo time for the different echo experiments. We have updated Figure 5 to reflect the correct filter-function peak position, and the updated noise magnitudes and exponents.

We have added a section to the supplement to describe the effect of finite-duration refocusing pulses on CPMG experiments. We have developed a theoretical model that allows us to extract the noise spectrum even in the presence of finite-duration pi pulses. We have updated Figure 5 to include the correct noise spectral density estimates and the filter function peak positions. Most notably, the frequencies of the CPMG filter functions now cluster around 2 MHz. This is because the maximum repetition rate of our pi pulses is limited to about 5 MHz due to the pulse length. (We have also modified the claim in the abstract and elsewhere in the text to indicate that our noise measurements occur over nearly 12 decades in frequency, as opposed to more than 13 decades.)

We have also removed the fit to the data in Fig 4e, because the argument that $T_2^n \propto n^\gamma$ relies on the specific form of the filter functions, which change when the finite refocusing pulse duration is accounted for.

- Figure 4b seems to show a large spread in estimated spectral exponent as a function of J. What causes this, and is it meaningful?

We suspect that this is primarily caused by the fact that the fits to the amplitude decay curves are not very sensitive to the spectral exponent parameter [see e.g. Figure 3d in Dial et al., PRL **110**, 146804 (2013) and Figure 3a in Medford et al., PRL **108**, 086802 (2012)]. While there may indeed be an underlying mechanism causing these trends in the spectral exponent, we cannot speculate on their origin. Moreover, the strong agreement between the data and fit in Fig 4c corroborates the claim that $\beta = \bar{\beta} \approx 0.95$.

- In the caption for 4d, it is stated "data points that fall between the horizontal gray lines are used to calculate the underlying noise spectrum." Why isn't all of the data used in this fit?

Data points with values larger than 0.85 and smaller than 0.15 are especially susceptible to noise. In principle, for example, the amplitude decay to infinitely long evolution time carries information about the low-frequency noise. However, the projection noise associated with our experiment easily overwhelms the signal in this regime. Likewise, differences between the measured data and 1 at very early times also indicate the high-frequency noise of the qubit, but the experimental noise can obscure these differences. These bounds are also in agreement with previous CPMG experiments.

- Please comment on expected electrical noise. E.g. from control electronics and Johnson noise.

We estimate the Johnson noise in our setup to be approximately $3 \times 10^{-21} \text{ eV}^2/\text{Hz}$. This includes contributions from cold resistors, as well as room-temperature electronics. We have added this to Figure 5. The approximately white voltage noise from our arbitrary waveform generator is around $2 \times 10^{-7} \text{ V}/(\text{Hz})^{1/2}$. Converting this to chemical potential noise gives about $2 \times 10^{-19} \text{ eV}^2/\text{Hz}$.

Complete list of changes to the manuscript

Main Text Changes:

1. We modified the abstract and text to indicate that our noise spectroscopy measurements span nearly 12 orders of magnitude, instead of more than 13.
2. We have updated Figures 4a-c, and we have removed the fit from Figure 4e.
3. We have updated Figure 5 with updated spin-echo-extracted and CPMG-extracted noise values, and to reflect the expected voltage noise from the arbitrary waveform generator.
4. We have modified the discussion of spin-echo and CPMG measurements to account for the fact that we analyze our data taking the finite duration of the pi pulses into account.

Supplemental Material Changes:

1. We have added a paragraph to the end of section IX describing the FFT-based method use to generate the power spectrum from FID measurements and noting the agreement between this method and the method used to generate the data shown in Figure 5 of the main text.
2. We have rewritten sections V, VII, and VIII (now sections VI-VIII) in the supplement to discuss noise spectroscopy with finite duration refocusing pulses. These sections now include a derivation of the spin-echo decay function, an approximate derivation of the CPMG filter function, and a discussion of the error incurred in extracting noise spectral densities with finite duration refocusing pulses. We have added Figures S3 and S5.
3. We have reanalyzed the data in Figure S8 (now S10) to account for the finite duration pi pulse and have updated the plots.

REVIEWERS' COMMENTS

Reviewer #4 (Remarks to the Author):

These modifications and responses have mitigated my concerns, and I recommend this work for publication. I commend the authors once again on their careful study and well-written manuscript.